# Evaluation of the Phytoremediation Potential of the *Sinapis alba* Plant Using Extractable Metal Concentrations

**DOI:** 10.3390/plants12173123

**Published:** 2023-08-30

**Authors:** Nicoleta Vasilache, Elena Diacu, Sorin Cananau, Anda Gabriela Tenea, Gabriela Geanina Vasile

**Affiliations:** 1Faculty of Chemical Engineering and Biotechnologies, University Politehnica of Bucharest, 1-7, Polizu, 011061 Bucharest, Romania; elena_diacu@yahoo.co.uk; 2National Research and Development Institute for Industrial Ecology ECOIND, 57-73 Drumul Podu Dambovitei, Sector 6, 060652 Bucharest, Romania; gabriela.vasile@incdecoind.ro; 3Faculty of Mechanical and Mechatronic Engineering, University of Science and Technology Politehnica Bucharest, 313, Splaiul Independentei, 060042 Bucharest, Romania; s_cananau@yahoo.com

**Keywords:** phytoremediation, bioaccumulation coefficient, response surface methodology, PCA analysis, regression multiple

## Abstract

Testing the feasibility of soil phytoremediation requires the development of models applicable on a large scale. Phytoremediation mechanisms include advanced rhizosphere biodegradation, phytoaccumulation, phytodegradation, and phytostabilization. The aim of this study was to evaluate the phytoremediation potential of the *Sinapis alba.* Identification of the factors influencing the extraction process of metals from contaminated soils in a laboratory system suitable for evaluating the phytoavailability of these metals in three solutions (M1-CaCl_2_, M2-DTPA, and M3-EDTA) included the following: distribution of metals in solution (Kd), soil properties and mobile fractions (SOC, CEC, pH), response surface methodology (RSM), and principal component analysis (PCA). The evaluation of the phytoremediation potential of the Sinapis alba plant was assessed using bioaccumulation coefficients (BACs). The accumulation of heavy metals in plants corresponds to the concentrations and soluble fractions of metals in the soil. Understanding the extractable metal fractions and the availability of metals in the soil is important for soil management. Extractable soluble fractions may be more advantageous in total metal content as a predictor of bioconcentrations of metals in plants. In this study, the amount of metal available in the most suitable extractors was used to predict the absorption of metals in the Sinapis alba plant. Multiple regression prediction models have been developed for estimating the amounts of As and Cd in plant organs. The performance of the predictive models generated based on the experimental data was evaluated by the adjusted coefficient of determination (aR2), model efficiency (RMSE), Durbin–Watson (DW) test, and Shapiro–Wilk (SW) test. The accumulation of the analyzed metals followed the pattern Root > Pods > Leaves > Seeds, stems > Flowers for As and Leaves > Root > Stem > Pods > Seeds > Flowers for Cd in soil contaminated with different metal concentrations. The obtained results showed a phytoremediation potential of the *Sinapis alba* plant.

## 1. Introduction

Plants are recognized for their ability to absorb minerals from the soil, but alongside the beneficial ones, there are also heavy metals. The development of the metal and oil extraction industry, the use of pesticides, and environmental pollution constitute, however, a danger regarding the safety of consuming plants from uncontrolled environments. Metals persist in the soil for a long time, having the ability to be transferred in the food chain [1,2], and the evaluation of their content in the soil and the estimation of their transfer rates to the vegetation present great interest [3]. The type of metals present and their concentration decide if contaminated soil needs to be treated. Sources of harmful metal pollution must greatly increase the amount of soil metals available to plants before they can be recognized. The solubility of the metal linked with the solid phase has the most impact on how easily dangerous metals are absorbed by plants in the soil [4,5,6]. The absorption, mobility, and toxicity of the metal to plants and animals that consume it are significantly determined by the degree of dispersion of soluble species. Factors influencing the solubility and distribution of chemically available metal species in soil influence the kinetics of sorption–desorption reactions, the metal concentration in the extraction solution, and the form of soluble or insoluble chemical species [7,8]. An accurate estimate of the phytoavailability of metals in soil and solid waste is becoming an increasingly serious risk. The evaluation of metal concentrations and their distribution can be associated with the understanding of the phytoavailability process of the metal. These attributes can be evaluated from the soil matrix using analytical techniques (spectroscopy) or approaches that address single chemical extractions of the metals. Biological instruments supply information regarding the direct entry of the contaminant into the plant, including measuring the plant’s response to its toxicity.

Predicting the phytoavailability of metals in natural or accidentally polluted soils with hazardous metals is relatively difficult mostly due to the variety of soil types. The main factor determining the phytoavailability of dangerous metals in soil is the solubility of the metal paired with the solid phase [9]. The rate of dissolution of soluble species substantially impacts the amounts of absorption, mobility, and toxicity of the metal in plants. Factors influencing the dissolution and distribution of chemically soluble metal species in soil include soil characteristics (metal concentration, mineralogy, particle size distribution) and soil processes [10]. They influence the kinetics of sorption–desorption reactions, the metal concentration in the extraction solution, and the form of soluble or insoluble chemical species [11]. The assessment and prediction of toxic metals in soil requires knowledge of their distribution in the solid–liquid system, expressed by the distribution coefficient Kd [12,13,14,15]. The Kd value of metal indicates the effect of various reactions in the solid and liquid phases. Kd values based on the soluble phase are often correlated with soil properties. Due to the fact that part of the metal in the solid phase is quickly exchanged with the solution phase, different extraction techniques were used to quantify the mobile phase. Phytoavailability of metals is estimated by different chemical extraction methods such as neutral salts, weak acids, organic extraction, and resin-based techniques, all with varying success [16,17,18]. Systematic studies on the distribution of toxic metals in contaminated soil samples were carried out using different extraction solutions, including EDTA [19,20,21,22]. Other studies have tried quantitative prediction of phytoavailability using extraction techniques [23,24,25,26,27,28,29].

In this study, an analysis of the extraction capability of several extraction procedures was utilized to predict the phytoavailability of the metals As and Cd in contaminated soils. *Sinapis alba* was selected to investigate the accumulation potential of As and Cd due to its capacity to retain several potentially hazardous elements [30].

The objective of this study was to analyze the potential of the Sinapis alba plant for phytoremediation of the As and Cd from polluted soils of the plant and to develop prediction models for predicting the metal concentrations of this plant using the extractable metal concentration.

## 2. Results

### 2.1. Characterization of Soils and Metallic Mobile Fractions

To understand the transport and bioavailability of extractable metals, we performed a detailed experimental study on the determination of the extractability of some solutions using the distribution coefficients Kd for As and Cd in contaminated soil samples using the laboratory batch method. Two crop soil samples taken from different areas (Soil1, Soil2) were first mixed with amendments to improve physical properties such as filterability, particle structure, and water infiltration necessary for optimal plant growth and then enriched with different concentrations of As, Cd, and Ni. Ni concentration was added in order to simulate multiple contaminations. The mobility process of Ni in the soil was not analyzed in the present study. The main characteristics of the initial soil samples are shown in Table 1.

In this study, the spiking method was used to obtain soil samples contaminated with different concentrations of As, Cd, and Ni. The total concentrations of As and Cd in the artificially contaminated soil samples are presented in Table 2 for Soil_1_ and Soil_2._ Metal analysis was performed with the ICP-EOS method according to the international standards.

The mobility of As and Cd metals in contaminated soils was investigated using three types of extraction solutions.

The extracting solutions used in the experiment were as follows:M-DIwater—1:10 ratio soil: extraction solution (M);M1-CaCl2—0.002 mol/L Diethylene triamine penta-acetic acid, 0.01 mol/L Calcium chloride, 1:5 ratio soil: extraction solution (M1);M2-DTPA—0.01 mol/L Calcium chloride, 0.1 mol/L Triethanolamine, 0.005 mol/L Diethylene triamine penta-acetic acid, 1:10 ratio soil: extraction solution (M2);M3-EDTA—1 mol/L Acetic acid, 0.01 mol/L Ethylene diamine tetra-acetic acid disodium salt, 1:10 ratio soil: extraction solution (M3).

The affinity of metals to the soil surface was also reflected by the distribution coefficient (Kd), which shows the ability of soil to retain the metal in the solid phase or release it into the extraction solution. Table 3 shows the values of As concentrations and the values of the Kd of As in all studied extraction solutions, while the values of Cd concentrations and associated Kd values in the studied extraction solutions are presented in Table 4.

### 2.2. Factors Influencing the Phytoavailability of Metals in Soils Contaminated with As and Cd

Soil pH strongly affects the speciation and mobility of metals in both soil and soil solution. The pH values of the extraction solutions are presented in Table 5.

CEC is used as an indicator of soil nutrient retention capacity. The organic carbon in the soil is a natural or anthropogenic dynamic component, and due to the decomposition of the organic matter in the soil, the impact it has on it is significant. Table 6 shows the cation exchange capacity values estimated in the extraction solutions and the organic carbon values in the contaminated soil samples.

### 2.3. Evaluation of the Influence of Factors on the Extraction of Metals As and Cd from Contaminated Soils Using PCA and RSM

In order to understand the complex relationship between the characteristic parameters of the soil and the distribution of the metal in the extraction solution, principal component analysis (PCA) was used. This technique reflects how much each variable contributes to the correlation of the data and the interpretation of the relationship between the variables [31]. The PCA analysis generated two main components. To evaluate each variable and how it affects the response variable Kd, the response surface methodology (RSM) was used. The response surface and contour plots of KdAs and KdCd were generated based on the obtained results [32]. Figure 1 shows the 3D and contour plots of the response surface of the KdAs distribution coefficients in the extraction solutions M1 (a), M2 (b), and M3 (c) under a single 10 mg/kg As contamination (S3).

The values of the principal components after Varimax rotation for As extraction solutions (M2, M2, M3) are presented for S3 in Table 7, for S4 in Table 8, for S5 in Table 9, and for S6 in Table 10.

The response surface and contour plots of KdAs in the extraction solutions M1 (a), M2 (b), and M3 (c) in the case of increasing the contaminant dose to 25 mg/kg As (S4) are presented in Figure 2.

The response surface and contour plots of KdAs in the extraction solutions M1 (a), M2 (b), and M3 (c) in the case of contamination with As 15mg/kg and Cd 3 mg/kg (S5) are presented in Figure 3. Figure 4 shows the response surface and contour plots of the KdAs in the extraction solutions M1 (a), M2 (b), and M3 (c) in the case of contamination with As 15 mg/kg, Cd 3 mg/kg, and Ni 10 mg/kg (S6).

In the same way, we evaluated the evolution of the Cd desorption process in the three types of extraction solutions in the representative soils of this study.

The values of the principal components after Varimax rotation for Cd extraction solutions (M2, M2, M3) are presented for S1 in Table 11, for S2 in Table 12, for S5 in Table 13, and for S6 in Table 14.

Figure 5 shows the response surface and contour plots of KdCd in the extraction solutions M1 (a), M2 (b), and M3 (c) in the case of a single contamination with Cd 2 mg/kg (S1).

The response surface and contour plots of KdCd in the extraction solutions M1 (a), M2 (b), and M3 (c) in the case of single contamination with Cd 5 mg/kg (S2) are presented in Figure 6. The matrices of Pearson coefficients and the 3D and contour plots of the KdCd in the extraction solutions M1 (a), M2 (b), and M3 (c) in the case of contamination with As 15 mg/kg and Cd 2 mg/kg (S5) and of combined contaminations with Cd 3 mg/kg, As 15 mg/kg, and Ni 10 mg/kg (S6) are presented in Figure 7 and Figure 8.

### 2.4. Evaluation of the Extraction Capacity of As and Cd from Contaminated Soils in M1-CaCl_2_, M2-DTPA, and M3-EDTA Solution Comparison with Water Extraction

The diversity of influencing factors of the sorption/desorption process of toxic metals in soils can make it difficult to derive some reference values of Kd distribution coefficients of toxic metals to be used as reliable indicators for evaluating the remediation process of contaminated soils. A constant update of the Kd values of heavy metals may be necessary for a thorough understanding of the influence of different factors on these processes and the determination of the extraction capacity of the solutions. Figure 9 and Figure 10 show the variation in the distribution coefficient Kd in the three extractive solutions, M1-CaCl_2_, M2-DTPA, and M3-EDTA for As and Cd, respectively, in soils contaminated with different concentrations of metals.

### 2.5. Evaluation of the Accumulation Potential of As and Cd in the Sinapis alba Plant

The bioaccumulation coefficient (BAC) was found as the ratio of the metal concentration in the plant organs to the metal concentration in the soil. BAC shows the ability of plants to retain toxic metals. The results obtained from the analysis of As and Cd concentrations in the parts of *Sinapis alba* grown in the greenhouse in soils contaminated with different metal concentrations (As, Cd, and Ni), as well as the accumulated metal concentrations in the plant, are presented in Table 15. The BAC-Cd and BAC-As values determined in the root (R), stem (St), leaves (L), flowers (F), pods (P), and seeds (Se) of *Sinapis alba* grown in soils contaminated with As, Cd, and Ni (S5 and S6) and the metal accumulation pattern in the plant are shown in Table 16.

### 2.6. Predictive Models Useful in Evaluating the Concentration of As and Cd in Sinapis alba Plant

#### 2.6.1. Evaluation of the Interdependence of the Variable Parameters Used in the Development of Multiple Regression Models for the Evaluation of As and Cd Concentrations in the *Sinapis alba*

PCA analysis was used to statistically identify the relationship between the extractable metal concentration as values of the KdAs and KdCd distribution coefficients obtained in the solution with the highest effectiveness for these metals (M1), the metal concentrations in the plant organs, and the metal concentration in the whole plant. The PCA analysis was applied to the soil contaminated with As and Cd (S5). The obtained results are presented in Table 17

#### 2.6.2. Prediction Models of As and Cd Concentrations in the *Sinapis alba*

In this study, multiple regression models for predicting the concentration of As and Cd in the organs of the plant *Sinapis alba* grown in the laboratory greenhouse were developed using the extractable metal concentrations evaluated by the distribution coefficients KdAs and KdCd in the analyzed solutions and the accumulation of the concentration of metal in the plant. The equations of the prediction model, the average experimental and predictive values of the metal in the plant organs, and the values of the parameters used in the validation process of the multiple regression model (aR^2^, DW, SW, RMSE) are presented in Table 18 for As and in Table 19 for Cd (N is the number of observations and k is the number of variables)

## 3. Discussion

### 3.1. Evaluation of the Influence of Factors on the Extraction of As and Cd from Contaminated Soils Using PCA and RSM

The sorption process is associated with processes such as ion exchange, precipitation, adsorption, and complexation by which dissolved toxic metal ions can bind to the soil causing metal accumulation. Desorption includes the extraction of metal ions absorbed from the soil into the soil–solution system, spreading the contamination over a large surface [33]. The extraction capacity of the three extraction solutions (M1-CaCl_2_, M2-DTPA, and M3- EDTA) was evaluated using the extractable As and Cd concentrations as values of the distribution of the metals analyzed in the solution phase, KdAs and KdCd. EDTA and DTPA are two chelating agents commonly used in sorption–desorption studies due to their ability to form stable complexes with a wide variety of metals. The nonselective nature of EDTA in metal extraction is a disadvantage, as this agent extracts a wide variety of metals, including alkaline earth cations such as Ca and Mg. Another disadvantage would be that EDTA is hardly biodegradable and can remain adsorbed on soil particles [34,35]. With calcium being the primary cation of the adsorption complex of soils, the M2-CaCl_2_ solution is more capable of extracting other cations adsorbed on the surface of soil particles than other solutions without Ca. Due to CaCl_2_, the Ca-DTPA complex facilitates ion exchange by forming complex combinations with various forms of toxic metals [36].

The evaluation of the factors that can influence the desorption process of As and Cd in the three extracting solutions was determined using PCA and the response surface plots of KdAs and KdCd. The experimental data obtained proposed the correct evaluation of the concentrations of toxic metals in soils with a high level of contamination. The aqueous extracts were not used in this study, because the water used as an extraction solution extracts soluble forms of the chemical species of As and Cd that do not reflect the real level of contamination of the analyzed soils. The experiment focused on identifying an extractant (for As and Cd) designed to ensure the maximum efficiency of the desorption process in contaminated soils, similar to other studies carried out [37].

Soil pH is an important factor influencing the availability of metals in the soil. Previous studies have established a relationship between soil pH and metal availability [32]. The desorption of metals from the soil particle structure is also favored by the low pH value of the extraction solution. Various studies confirm the dependence of metal mobility on pH value [38]. The cation exchange capacity (CEC) of the soil represents the quantity of cations that can be retained on the clay-humic adsorption complex of the soil, at a certain pH. CEC is used as a measure of soil fertility, indicating the ability to retain nutrients in the soil. The richer the soil in clay and organic matter, the more important the CEC is [39]. The cation exchange capacity is related to the sum of the exchangeable bases and can be calculated empirically by summing the basic exchangeable cations (Ca^2+^, Mg^2+^, Na^+^, K^+^) and the acidic cations (H^+^, Al^3+^, NH_4_^+^) that are attracted to the negative charges on the particle surface of the soil [40]. The mobility of As, tested on different types of clay soils, indicated weak surface retention of minerals in the composition of these types of soils [36]. Data from literature studies show that Cd desorption increases at pH ≈ 2, with Cd uptake being negatively correlated with pH value [41]. Of particular importance, in the context of finding metal concentrations, is the retention potential of metals in the surface layer of soils. This has important implications for the mobility of metals in the soil profile (depth) and the bioavailability of metals to organisms living in the surface layer and plants. Crop soils generally have low SOC concentrations. Clay soil types generally retain more organic matter than sandy soils and therefore more organic carbon [42]. Changes in stable SOC usually occur very slowly (over decades) and, therefore, changes in the organic carbon content of agricultural soils are small, being most often determined in the upper layers. Due to its stability over time, SOC can be an important factor influencing the actual sorption/desorption processes of different toxic metals in soil [43,44].

Numerous studies have shown the influence of pH on the behavior of Cd and As mobility. In the aqueous environment, Cd^2+^ shows relative mobility, with the process depending on the pH, the presence of organic molecules, and the hardness of the water. The concentration of Cd in the soil can reach high levels of total concentrations, but the desorption of Cd^2+^ ions and its absorption by plants is supported by an increased acidity [45].

The study of As sorption/desorption processes in soil indicated an increase in the desorption process at higher concentrations explained by the non-selective sorption to the surface of soil particles of specific As chemical species. Similar studies showed the influence of pH in increasing the mobility of high concentrations of As due to its association with Fe ions [46].

The sorption/desorption distribution coefficient (Kd) is an important parameter frequently studied to understand the mobility of a compound in the environment and its distribution between water, sludge, soil, and sediment compartments. In addition, Kd is an essential parameter for evaluating the bioavailability and leaching processes; therefore, it is directly related to distribution coefficients. The desorption process involves selecting a distribution coefficient with the lowest value (high mobility) [47]. The characterization of Kd in the laboratory system and the extrapolation in the real environment can be difficult, mostly due to the complexity of the mechanisms involved in the sorption/desorption process [43].

PCA was applied to identify the variability of factors that influence the desorption process of As and Cd in the soluble phase and which can influence a correct assessment of metal concentrations in contaminated soils. The multivariate data set consisted of five variables: two basic characteristic parameters of the soil (SOC, metal concentration in the soil) and three characteristic parameters of the extraction solutions (CEC, pH, and Kd). In order to eliminate errors due to non-correlation and maximize the variance of the components of the two factors, the correlation matrix extracted by PCA was subjected to orthogonal Varimax rotation. The significant factors obtained after Varimax rotation and loadings indicate how each parameter is related to these factors [48,49].

The next applied step included the application of RSM for the simultaneous evaluation of the relationship between the effects of the parameters involved in the sorption–desorption process of mobile As and Cd species [50]. The data obtained were aimed at evaluating the extraction capacity of the solutions used in this study. The response surface and contour plots of the distribution coefficients KdAs and KdCd were used to follow the evolution of the extraction process and of the interdependence reactions determined by the variability of the characteristic parameters of the soil and of the extraction solutions. Rapid evolution of the interaction effects is shown by the close-curved lines in the contour plot [45].

#### 3.1.1. The Desorption Process of As in Contaminated Soils

In the case of S3 soil samples contaminated with As, the results show that the metal extraction from the soil is moderately influenced by the SOC variation (PC1_M1: +0.642; PC1_M2: −0.475; PC2_M3: −0.720) and the CEC variation for M3. Increasing the contamination level reduces the extractable metal concentration in solutions M1 and M2 (Table 7). The extraction of As from the S4 soil is strongly positively influenced by the CEC variability in the M1 and M3 solutions (PC2_M1: −0.916 and PC2_M3: 0.733) and by a low concentration of SOC in the soil for the M2 solution. Increasing the concentration of As in the soil decreases the efficiency of the metal desorption process in the solution (Table 8). As concerns desorption from soil contaminated with As and Cd, S5 is favored by low pH values in solutions M1 and M2 (PC1_M1: 0.598; PC2_M2: −0.896) and by CEC variability in solution M3 (PC2_M3: 0.800). The metal concentration in the soil has a negative influence on the extraction of As in the analyzed solutions (Table 9). The pH was the factor that moderately influenced the extraction of As from soil contaminated with As, Cd, and Ni (S6) in the three solutions (PC2_M1: 0.634; PC2_M2: −0.540; PC2_M3: −0.545). Low pH values favor the desorption of As in solutions M1 and M3. Another factor with moderate variability was SOC for solutions M1 and M2 and CEC for solution M3 (Table 10).

The response surface and contour plots of KdAs presented that the As sorption/desorption process is accompanied by interactions between the parameters determined as variable factors that influence the extraction of the metal from the S3 soil (Figure 1a–c) for all three extraction solutions. The increase in the level of contamination in the S4 soil slightly stabilizes the evolution of the desorption process, but at concentrations of As higher than 26 mg/kg in the soil, the metal is retained in the solid fraction (Figure 2a–c). The extraction of As from soils S5 and S6 (Figure 3 and Figure 4a–c) indicated a reduction in the metal desorption process in the extraction solutions, also confirmed by the increase in KdAs.

#### 3.1.2. The Desorption Process of Cd in Contaminated Soils

The results obtained in the case of soil sample S1 contaminated with Cd show that the extraction of the metal from the soil is strongly influenced by the variability of the pH in solution M1 (PC2_M1: −0.798), CEC in solutions M2 and M3 (PC2_M2: −0.891; PC2_M3: −0.892), and low values of metal concentrations in soil (Table 11). The extraction of Cd from the S2 soil in the M1 solution is influenced by the variability of the metal concentration and SOC in the soil (PC2_M1: −0.695). In the M2 solution, the variable factors that can influence the Cd desorption process are the CEC of the extraction solution (PC2_M2: −0.696) and the reduced level of contamination. In solution M3, Cd extraction is moderately influenced by pH and SOC (Table 12). The desorption of Cd from the soil contaminated with As and Cd (S5) is favored by the low values of the metal concentration in the soil and by CEC in the M2 and M3 solutions. For solution M1, low pH values favor the desorption process in the extraction solution (Table 13). The data obtained in the case of soil contaminated with As, Cd, and Ni (S6) show a strong influence of pH (PC1_M1: −0.972) and CEC (PC2_M1: −0.958) in the extraction process of Cd in solution M1. The extraction of Cd in the M2 solution is influenced by the pH variation (PC1_M2: 0.904) and the metal concentration in the soil (PC2_M2: −0.923). The extraction of Cd in solution M3 is favored by low concentrations of CEC (PC2_M3: −0.914) and the contamination level (PC1_M3: −0.926) of the soil (Table 13).

The response surface and contour plots of KdCd showed a slow evolution of the Cd desorption process in S1 soil (Figure 5a–c) for all three extraction solutions. Increasing the concentration of Cd to 5 mg/kg in soil S2 (Figure 6a–c) had the effect of stabilizing the metal sorption/desorption process over time. The extraction of Cd from soils S5 and S6 (Figure 7 and Figure 8a–c) indicated a slowing down of the desorption process accompanied by interactions between the characteristic parameters of the soil and the extraction solutions determined as factors that strongly influenced the distribution of the metal in the analyzed solutions.

### 3.2. Evaluation of the Extraction Capacity of the Studied Solutions for Determining the Extractable As and Cd Concentrations Compared to the Extracting of Water from Contaminated Soils

To determine the concentrations of As and Cd extracted from soils contaminated with different concentrations of As, Cd, and Ni, three types of extractive solutions (M1, M2, and M3) were used in comparison with deionized water (DIwater-M) extract. The evaluation of the extraction capacity of the determination of extractable metal concentrations was found using the values of the KdAs and KdCd distribution coefficients for these metals in the analyzed solutions (Figure 9 for As and Figure 10 for Cd). The obtained results determined the following order of evaluation of the effectiveness of the analyzed extraction solutions:

As extraction: S3: EDTA > CaCl_2_ > DTPA > DIwater; S4: EDTA > CaCl_2_ > DTPA > DIwater; S5: CaCl_2_ > DTPA > EDTA > DIwater; S6: CaCl_2_ > EDTA > DTPA > DIwater.

Cd extraction: S1: EDTA > CaCl_2_ > DTPA > DIwater; S2: EDTA > CaCl_2_ > DTPA > DIwater; S5: CaCl_2_ > DTPA > EDTA > DIwater; S6: CaCl_2_ > EDTA > DTPA > DIwater. Other studies have obtained similar results [51].

The experimental results showed that M1-CaCl_2_ and M3-EDTA are solutions that can best evaluate the concentration of toxic metals in soils contaminated with As and Cd. The method of extracting As and Cd metals into DIwater was not adequate for determining the amounts of extractable metal from soils with high levels of contamination.

### 3.3. Evaluation of the Accumulation Potential of As and Cd in the Sinapis alba Plant

Most plants growing in soils polluted with toxic metals show a physiology in which they can avoid the uptake of metals, while in others, the accumulation process may differ between different parts of the plant. Mechanisms of toxic metal accumulation at the whole plant level involve the regulation of several processes, including metal uptake by the root [52]. Accumulation of toxic metals in plant roots can cause disturbances in the ratio of nutrients in plant tissues and changes in water balance [53]. To assess the bioaccumulation potential of *Sinapis alba*, As and Cd concentrations were analyzed in the root, stem, leaves, flowers, pods, and seeds of plants grown in soils contaminated with As, Cd, and Ni. The total metal concentration in the plant was obtained by summing the concentrations in the plant organs. The evaluation of As and Cd concentrations (Table 15) in the plant grown in S5 and S6 soils indicated the accumulation of As in the root (2.56 ± 0.12 mg/kg in S5 and 9.02 ± 0.12 mg/kg in S6) and the accumulation of Cd in leaves (1.46 ± 0.10 mg/kg in S5 and 2.52 ± 0.10 in S6).

The BAC-As and BAC-Cd bioaccumulation coefficients (Table 16) showed values between 0.1 and 1 in the two soils S5 and S6. The highest average values of BAC-As obtained (0.166 for soil S5 and 0.851 for S6) showed a moderate potential for accumulation of As in *Sinapis alba* root. The highest mean BAC-Cd values obtained (0.567 for soil S5 and 0.813 for S6)) indicated a moderate potential for phytoaccumulation of Cd in *Sinapis alba* leaves. The accumulation pattern of As and Cd metals in the plant parts was in the following order: for AsS5—Root >Pods > Leaves > Stem, Seeds >Flowers; for AsS6—Root > Leaves > Stem, Flowers > Pods, Seeds; for CdS5—Leaves > Root > Stem > Pods > Seeds > Flowers; and for CdS6—Leaves > Root > Stem >Flowers > Pods > Seeds.

Other studies confirm the retention of As predominantly in the roots of *Sinapis alba* [54] and translocation of Cd in the aboveground part of the plant [30,55]. The results suggest that the ability of the *Sinapis alba* plant to survive high concentrations of As and Cd indicates that it could be used in a phytoremediation strategy for contaminated soils in areas affected by the mining industry.

### 3.4. Predictive Models Useful in Evaluating the Concentration of As and Cd in Sinapis alba Plant

The accumulation of heavy metals in plants is connected to the concentrations and chemical fractions of metals in the soil. Understanding dissolved chemical elements and the availability of metals in the soil is essential for soil management. Extractable fractions may be more advantageous in total metal content as a predictor of metal bioconcentrations in plants. In this study, the amount of metal available in the most suitable extractors was used to predict the absorption of metals in the Sinapis alba plant.

#### 3.4.1. Evaluation of the Interdependence of the Variable Parameters Used in the Development of Multiple Regression Models for the Evaluation of As and Cd Concentrations in the *Sinapis alba* Plant

Principal components regression (PCR) is a regression technique similar to multiple linear regression that models the relationship between an original variable and predictor variables, using principal components instead of predictor values. The PCA analysis was used to evaluate the interdependence relationships between the parameters proposed for the development of predictive models of As and Cd concentrations in the Sinapis alba plant. The results of the PCA analysis for As showed that the first main component, PC1, is correlated with six variables (Kd_M1S5, Asplant_S5, Root_S5, Stem_S5, Leaves_S5, Pods_S5) varying together. The first main component correlates strongly with Asplant_S5 (correlation 0.933). This shows that the increase in the concentration of As in the whole plant determines the increase in the metal concentration in the Stem (correlation 0.920), Leaves (correlation 0.886), Root (correlation 0.680), and the decrease in the KdAs distribution coefficient (correlation −0.715). The second main component, PC2, correlates with the variables Flowers_S5 and Seeds_S5 but does not present an element of interest for the purpose of this evaluation. The results of the PCA analysis for Cd showed that the first main component, PC1, is correlated with six variables (Root_S5, Stem_S5, Flowers_S5, Pods_S5) varying together. The second main component, PC2, correlates strongly with three variables (Kd_M1S5, Asplant_S5, Root_S5, Leaves_S5). The PC2 component correlates strongly with Asplant_S5 (correlation 0.931), and the increase in this variable causes the increase in As concentration in the Leaves (correlation 0.812) and positively influences the KdCd values (correlation 0.763).

The obtained results confirmed the interdependence relationship between the variables proposed to be used in the development of prediction models of As and Cd concentrations in the Sinapis alba plant using extractable metal concentrations, evaluated with the KdAs and KdCd distribution coefficients.

#### 3.4.2. Prediction Models of As and Cd Concentrations in the *Sinapis alba* Plant

Regression modeling is one of the most widely used statistical processes for estimating relationships between dependent and independent variables, frequently applied in a wide range of successful applications. Multiple regression includes many techniques for modeling and analyzing variables to identify real-world problems. The conventional method is based on the assumption that the maximum accuracy of inaccessible data is obtained from models with the least amount of error in modeling the available data [56]. Regression models can be useful in monitoring the accumulation of toxic metals in plants grown in contaminated soils [57] or in assessing the phytoremediation potential of plants [58]. In this stage of the study, predictive models of As and Cd concentration values in the plant parts were developed which, according to the data obtained from the PCA analysis, present statistical relationships as variables.

The aR^2^ values obtained in the prediction models of the As concentration in the organs of the *Sinapis alba* grown in the S5 soil presented values between 0.9912 and 0.9974. The aR^2^ values obtained in the prediction models of the As concentration in the organs of the *Sinapis alba* grown in the S6 soil presented values between 0.9862 and 0.9989. The aR^2^ value was 0.9986 for the prediction models of the Cd concentration in the leaves of the *Sinapis alba* grown in the S5 soil and 0.9987 for the prediction model of the Cd concentration in the leaves of the plant grown in the S6 soil.

The Durbin–Watson test values for a significance level of 5%, used for the analysis of serial autocorrelation in the multiple regression models generated for the prediction of KdAs and KdCd (N = 12; k = 4), were between 1.85 and 2.66. The values of the Durbin–Watson test used for the analysis of serial autocorrelation in the gender-error multiple regression models for predicting the concentration of As and Cd in the parts of the *Sinapis alba* (N = 12; k = 2) were between 2.27 and 2.80. The results of the Durbin–Watson test indicate values higher than d_U_ (d—Durbin–Watson critical values—95%, d_L_—lower limit value, d_U_—upper limit value), so there is no autocorrelation between the residual values. The critical values for the models without intercept are d_L_ = 0.397 and d_U_ = 1.682 (N = 12, k = 4) and d_L_ = 0.674 and d_U_ = 2.268 (N = 12, k = 2) [59,60].

The results of the SW test for checking the normality of the data indicated no non-normality values for the independent variables KdAs, KdCd, and the concentration of As and Cd in the organs of *Sinapis alba* for a significance level of 5% (*p*-value < 0.05). The root mean square error, RMSE, values obtained in the prediction models showed values between 0.152 and 0.524 for KdAs and between 0.148 and 0.643 for KdCd. The RMSE values obtained in the prediction models showed values between 0.037 and 0.331 for the As concentration and values between 0.092 and 0.103 for the Cd concentration in the organs of Sinapis alba. Small RMSE values indicated a good fit of the models to the original experimental data sets.

Studies to date have sought different extraction methods that more effectively reflect the bioavailability of metals to be extracted from the soil–plant system. L. P. Gough developed multiple regression models for the prediction of copper, iron, manganese, and zinc levels in plants using DTPA and EDTA soil extractants [61]. Due to the characteristics of the soil parameters, the total concentration of metals in the soil cannot fully indicate the behavior of the metal in its mobility process in the soil–plant system. The relative contributions of variables on sorption/desorption mechanisms are complex and time-consuming.

#### 3.4.3. Limitation

This study may have possible limitations. Prediction models (stationary experimental design) of metal concentrations in *Sinapis alba* were developed using the best-fit extraction method of mobile metal concentration. The practicability of using the future results obtained is reduced to the area where the variables present values in the uncertainty range established by the experimental design and the time that determines the maturity of the plant.

## 4. Materials and Methods

### 4.1. Characterization of the Soil Samples and the Plant

Two uncontaminated topsoil samples were collected from different areas at a depth of 0–30 cm, dried at room temperature, and passed through a 2 mm sieve. The soil samples were mixed with universal soil amendment in a ratio of 1:3 to provide the plants with the nutrients required for rapid growth. The physical-chemical parameters characteristic of the soil were analyzed before the enrichment with metals. The method of enrichment of soil samples according to the experimental plan consisted of spraying with solutions of different and combined concentrations of metal in a repetitive thin layer, calculated by the amount of soil required for plant cultivation and leaching tests in three extraction solutions followed by homogenization [61,62]. After 10 days of stabilization, the six soil samples were homogenized and prepared according to the experimental design, as described in our previous work [55]. Four samples taken from each soil were used in the evaluation of the extraction capacity of three solutions (M1-CaCl_2_, M2-DTPA, and M3-EDTA). For comparison, deionized water was used [63]. The metal concentrations in the soil samples were determined after the stabilization of the contamination stage by mineralization in HNO_3_ and HCl in a volume ratio 1:3. The four soil–solution samples in the proportion specified by the experiment were stirred for 2 h at a frequency of 40 rotation/minute. Three samples were taken from the totally filtered leachates and the concentration of metals was analyzed. Four other samples from each soil were prepared for the cultivation of *Sinapis alba* seeds.

Germination of *Sinapis alba* seeds took place in the greenhouse system at a temperature of approximately 28 °C. Plants were grown in each sample lot (*n* = 4) corresponding to each contaminated soil type under laboratory conditions. For each sample, the sampling took place in two stages: in the first stage, the plants that reached the flowering stage were collected, and in the second stage, the plants that reached maturity were collected. The plant samples taken were dried by lyophilization (Lyophilizer Christ Alpha 1–2 LSCbasic, Osterode am Harz, Germany), separated according to the type of organ analyzed (root, stem, leaves, flowers, pods, and seeds), grinded (Mortar Grinder RM 200, Retsch Romania Verder scientific, Bucharest), and divided into three samples. Plant samples and seeds were taken before cultivation, were mineralized in a microwave digestion system (Ethos Up Microwave Digestion Systems—Milestone, Italy), and analyzed for metals of interest in the study. The obtained results indicated values of As and Cd concentrations in the seeds used before the experiment below the determination limit of the device. As and Cd metal analysis in *Sinapis alba* parts, extraction solutions, and soil were determined by inductively coupled plasma optical emission spectrometry (AVIO 500 ICP-EOS Hydride Generation Spectrometer FIAS 400 Perkin Elmer, USA). The characteristic parameters of the soil were determined as follows: pH and conductivity were determined electrochemically in aqueous extract 1:5; total nitrogen was determined volumetrically after mineralization in sulfuric acid and salicylic acid; total phosphorus was determined spectrometrically (Specord 210 Plus UV-Vis Spectrometer Analytik Jena, Germany) after mineralization in nitric acid and perchloric acid in a ratio of 1:5; soil organic carbon was determined by the Walkley–Black method by oxidation with potassium bichromate and sulfuric acid. The cation exchange capacity (CEC) was estimated empirically based on the content of cations in the extraction solutions with Equation (1):CEC = basic metal cations + acidic metal cations [meq/100g](1)
where basic metal cations represents the sum of the ions Ca^2+^, Mg^2+^, Na^+^, K^+^, Al^3+^, and NH_4_^+^.

The distribution coefficient Kd is a measure of the sorption of a metal as a metal ion in a geo-environment and is site-specific. It is a ratio between the amount of metal ions adsorbed on the soil mass and the amount of metal ions remaining in solution at equilibrium [64] and is calculated with Equation (2):Kd = Si/Ms [L/kg](2)
where Si is the metal concentration in the soil and Ms is the metal concentration in the extractive solution.

The bioaccumulation coefficient (BAC) is calculated (Equation (3)) as the ratio of metal concentration in the plant parts (roots, stems, leaves, flowers, and leaves) to those in the external medium, such as soil, to qualify trace metal accumulation in plants.
BAC = Cplant/Si(3)
where Cplant are trace metal concentrations in plant parts (mg/kg) and Si are trace metal concentrations in soil (mg/kg). Four categories of metal bioaccumulation are proposed: BAC value lower than 0.01 categorizes a plant as a non-accumulator, between 0.01 and 0.1 as a low accumulator, between 0.1 and 1.0 as a moderate accumulator, and between 1.0 and 10.0 as a high accumulator or hyperaccumulator [64].

### 4.2. Response Surface Methodology (RMS)

The response surface method (RSM) is a collection of mathematical algorithms and statistical techniques for building empirical models. Optimizing the response *Y* (output dependent variable) that is influenced by the independent variable *X* is carried out by carefully designing experiments that represent a series of tests, called runs. They modify the input variables in order to identify the reason for the change in the output response. The response can be plotted, either in three-dimensional space or as contour plots that help visualize the shape of the response surface. One of the main advantages of RSM is that a large amount of information can be obtained from a limited number of experiments. By constructing models and plots, the effects of variables and their interaction on the response can be studied. The response surface methodology uses statistical models. That is why practitioners must be aware that even the best statistical model is an approximation of reality. In reality, models and parameter values are unknown and subject to uncertainty. An estimated optimal point is not necessarily optimal in reality, due to estimation errors and model inadvertences. This method is quite difficult to build, requiring more time compared to other statistical models. However, the response surface methodology can help researchers to effectively improve their experimental studies. For example, Box’s initial response surface modeling allowed chemical engineers to improve a process that required expensive experiments [65,66].

### 4.3. PCA Analysis

In order to understand the complex relationship between the characteristic parameters of the soil and the statistical relationships between the variables of the multiple regression model developed in this study, PCA analysis was used [67]. The methodology used included the own analysis of the correlation matrix, the characteristic parameters of the soil, the extraction solutions, and the analyzed plants. Each variable shows a loading that shows how well a variable is accounted for by the model components. They reflect the contribution of each variable (parameter) to the significant variation in the data set and the relationship between the variables. The correlation was used due to the differences between the measurement units (metal concentration in mg/kg and Kd in L/kg) by normalizing the variables using division by their standard deviations. Varimax rotation was used to maximize the sum of variance of the squared loadings, as all coefficients can be either large or close to zero, with few intermediate values. This method simplified the interpretation of the PCA analysis results by associating each variable to a specific factor.

### 4.4. Multiple Regression Models

Multiple regression is a type of linear regression that extends the simple case of a dependent variable and an independent variable to several independent variables and allows for the estimation of coefficients of each independent variable, evaluating how well they explain the variation of the dependent variable. In addition, multiple regression can be used to test hypotheses about the interaction effects of different variables and to compare the fit of different models [68]. One of the main advantages of multiple regression is that it can include several independent variables, finding the complex nature of real-world phenomena. Multiple regression can identify interactions between variables, which can determine situations where the effect of one variable depends on the level of another variable. One of the disadvantages of multiple regression is that it can make the results difficult to interpret when multiple independent variables or complex interactions are involved. An essential condition in generating a predictable model is the verification of hypotheses and multiple regression conditions such as linearity, normality, independence, and multicollinearity. These conditions can be checked with diagnostic tests and charts. If these assumptions are violated, the results may be inaccurate or misleading. Overfitting can occur when too many independent variables are involved or when the variables are highly correlated with each other, and the model may lose its ability to generalize to new data.

Model evaluation shows the performance of the prediction model facilitating its practicability. The accuracy of a regression model indicates how close the predictive value is to the true value.

The methodology for the development of models for predicting the concentrations of As and Cd in the organs of the *Sinapis alba* plant followed the evaluation of the aR2 parameters, the Durbin–Watson test, the Shapiro–Wilk test, the probability value of each variable *p* < 0.05 included in the model, histogram plots, and the RMSE value.

The adjusted R square (aR2) determines the model fit of the dependent variables without considering the problem of overfitting when there are many variables and a complicated model. The adjusted R-square penalizes additional independent variables added to the model by adjusting the measure of error (RMSE or MAE) to prevent overfitting problems.

The root mean square error (RMSE) is the square root of the MSE calculated by the sum of the square of the prediction error, which is the experimental value minus the predicted value and then divided by the number of observations.

The Durbin–Watson test was used in the analysis of the autocorrelation of residuals indicating the influence of past values on future values in the data set generated by the model and was related to the tabulated critical values for the no-intercept model [69].

The Shapiro–Wilk test was used to check the normality of experimental data which works better with small data sets [70]. The test results did not reveal non-normality for the independent variable (*p*-value < 0.05 for a 5% significance level). Based on this result and after a visual examination of the histograms of the independent variables, it was decided to use the parametric Pearson correlation test.

Number Cruncher Statistical Systems (NCSS 2021) software was used for calculations and for statistical data analysis (PCA analysis, the response surface and contour plots, multiple regression equations and validation tests of the generated models, tests, plots, and calculations).

## 5. Conclusions

The research paper explores the results of using the Sinapis alba plant for soil phytoremediation, focusing on the bioaccumulation of the metals As and Cd. The study identifies the factors that influence the extraction of metals from contaminated soils and evaluates the extraction capacity of three solutions, M1-CaCl2, M2-EDTA, and M3-DTPA, and develops predictive multiple regression models to determine the concentrations of As and cd in the *Sinapis alba* plant using extractable metal concentrations. According to the study’s findings, As and Cd may be removed from contaminated soil by the Sinapis alba plant using phytoremediation.

The results obtained may be relevant in future research on the comparable effectiveness of EDTA- and CaCl_2_-assisted phytoremediation of soils contaminated with these metals using *Sinapis alba* [71].

## Figures and Tables

**Figure 1 plants-12-03123-f001:**
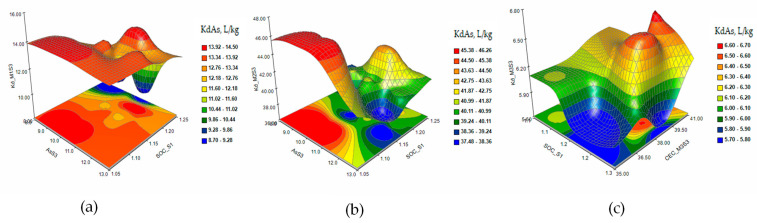
The response surface and contour plots for KdAs in extraction solutions M1 (**a**), M2 (**b**), and M3 (**c**) for a single contamination with 10 mg/kg As contamination (S3).

**Figure 2 plants-12-03123-f002:**
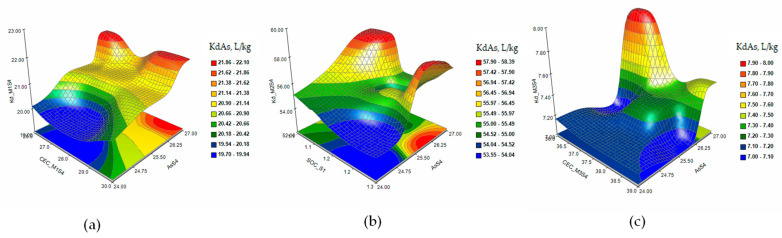
The response surface and contour plots of KdAs in extraction solutions M1 (**a**), M2 (**b**), and M3 (**c**) for a single contamination with 25 mg/kg As (S4).

**Figure 3 plants-12-03123-f003:**
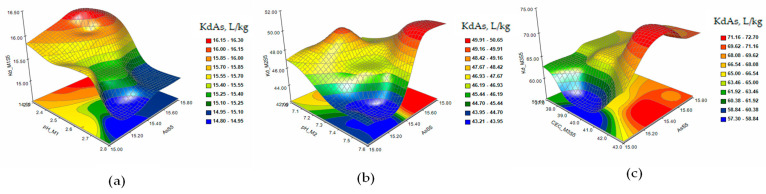
The response surface and contour plots of KdAs in the extraction solutions M1 (**a**), M2 (**b**), and M3 (**c**) in the case of contamination with As 15 mg/kg and Cd 2 mg/kg (S5).

**Figure 4 plants-12-03123-f004:**
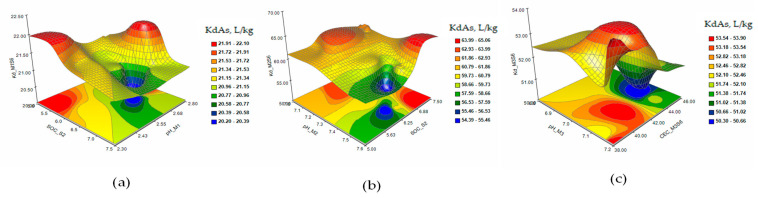
The response surface and contour plots of KdAs in the extraction solutions M1 (**a**), M2 (**b**), and M3 (**c**) in the case of combined contamination with As 15 mg/kg, Cd 3 mg/kg, and Ni 10 mg/kg (S6).

**Figure 5 plants-12-03123-f005:**
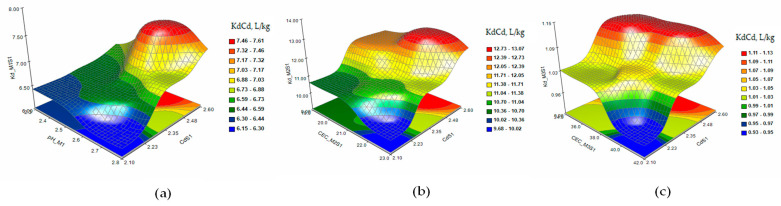
The response surface and contour plots of KdCd in the extraction solutions M1 (**a**), M2 (**b**), and M3 (**c**) in the case of single contamination with Cd 2 mg/kg (S1).

**Figure 6 plants-12-03123-f006:**
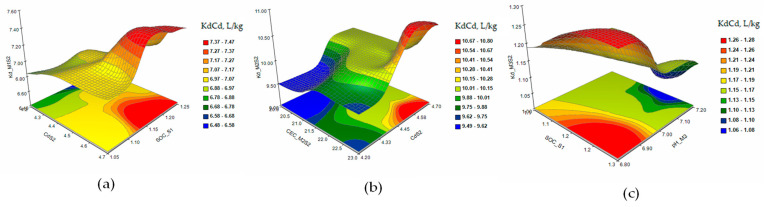
The response surface and contour plots of KdCd in the extraction solutions M1 (**a**), M2 (**b**), and M3 (**c**) in the case of single contamination with Cd 5 mg/kg (S2).

**Figure 7 plants-12-03123-f007:**
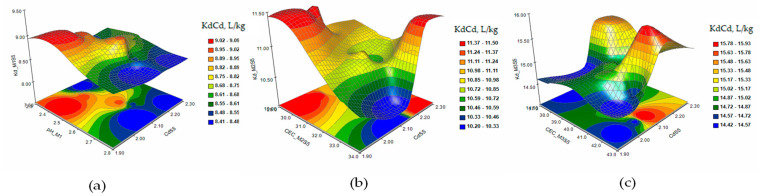
The response surface and contour plots of KdCd in the extraction solutions M1 (**a**), M2 (**b**), and M3 (**c**) in the case of contamination with Cd 2 mg/kg and As 15 mg/kg (S5).

**Figure 8 plants-12-03123-f008:**
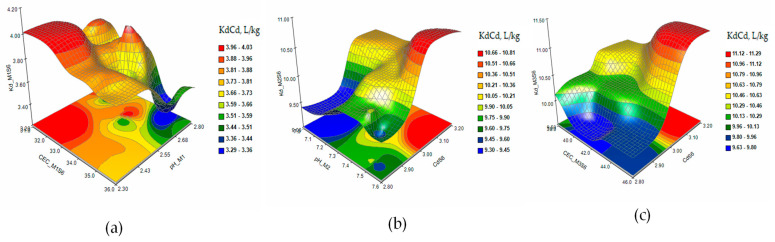
The response surface and contour plots of KdCd in the extraction solutions M1 (**a**), M2 (**b**), and M3 (**c**) in the case of combined contamination with Cd 3 mg/kg, As 15 mg/kg, and Ni 10 mg/kg (S6).

**Figure 9 plants-12-03123-f009:**
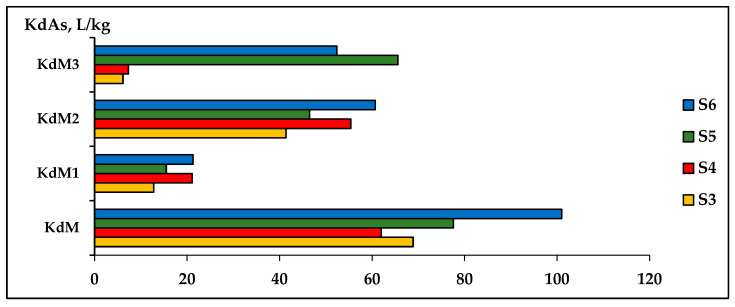
The variation in the distribution coefficient KdAs in three extracting solutions (M1-CaCl_2_, M2-DTPA, and M3-EDTA) evaluated comparably with water extraction (KdM) in S3, S4, S5, S6 soils.

**Figure 10 plants-12-03123-f010:**
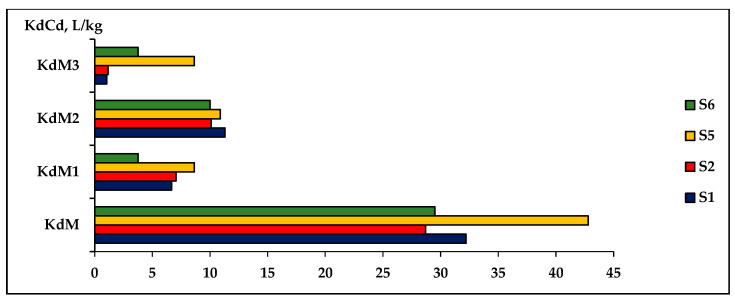
The variation in the distribution coefficient KdAs in three extracting solutions (M1-CaCl_2_, M2-DTPA, and M3-EDTA) evaluated comparably with the water extraction (KdM) in S1, S2, S5, S6 soils.

**Table 1 plants-12-03123-t001:** Characteristics of soils with amendment used in the experiment.

Parameter	UM	Soil1 (s1)	Soil2 (s2)
As	mg/kg	0.35 ± 0.06	0.56 ± 0.09
Cd	mg/kg	0.02 ± 0.002	0.07 ± 0.002
Ni	mg/kg	10.2 ± 0.16	9.96 ± 0.69
Zn	mg/kg	65.3 ± 0.87	85.1 ± 0.67
Cu	mg/kg	39.2 ± 0.24	26.9 ± 0.18
Co	mg/kg	6.22 ± 0.65	5.16 ± 0.87
Cr	mg/kg	13.2 ± 0.27	13.5 ± 0.25
Fe	mg/kg	14,652 ± 235	13,526 ± 166
Mn	mg/kg	609 ± 25	512 ± 21
Pb	mg/kg	11.3 ± 0.13	6.52 ± 1.02
Na	mg/kg	183 ± 1.87	234 ± 1.68
Ca	mg/kg	10,325 ± 215	12,135 ± 209
Mg	mg/kg	3465 ± 84	4135 ± 69
K	mg/kg	2132 ± 62	3025 ± 51
SOC	%	0.92 ± 0.35	6.42 ± 1.20
pH	pH unit	7.12 ± 0.31	6.93 ± 0.25
Conductivity	µS/cm	80.7 ± 5.6	115 ± 6.8
Ptot	mg/kg	3139 ± 102	4297 ± 265
Ntot	mg/kg	6582 ± 152	8320 ± 231

Mean value (*n* = 4) ± SD (standard deviation); Ntot—total nitrogen; Ptot—total phosphorus; SOC—soil organic carbon.

**Table 2 plants-12-03123-t002:** The concentration of As and Cd in different artificially contaminated soils.

Soil	AsSi *	CdSi *
S1 (Soil1 with Cd 2 mg/kg)	0.28 ± 0.02	2.32 ± 0.13
S2 (Soil1 with Cd 5 mg/kg)	0.51 ± 0.04	4.47 ± 0.52
S3 (Soil1 with As 10 mg/kg)	10.6 ± 1.21	0.12 ± 0.06
S4 (Soil1 with As 25 mg/kg)	26.1 ± 1.85	0.09 ± 0.04
S5 (Soil1 with As 15 mg/kg and Cd 2 mg/kg)	15.1± 1.65	2.12 ± 0.07
S6 (Soil2 with As 15 mg/kg, Cd 3 mg/kg, Ni 10 mg/kg)	16.9 ± 1.37	2.97 ± 0.32

Mean value (*n* = 12) ± SD; * AsSi/CdSi—metal concentration in soil (i = 1 ÷ 6) (mg/kg).

**Table 3 plants-12-03123-t003:** The values of As concentrations in the CaCl_2_, DTPA, and EDTA extraction solutions and the values of the Kd distribution coefficients of As in these solutions.

Parameter (As)	S3	S4	S5	S6
M	0.15 ± 0.02	0.42 ± 0.04	0.20 ± 0.01	0.15 ± 0.02
M_C_	1.20 ± 0.24	1.24 ± 0.01	0.99 ± 0.04	0.50 ± 0.05
M_D_	0.36 ± 0.03	0.47 ± 0.02	0.33 ± 0.02	0.18 ± 0.02
M_E_	2.43 ± 0.14	3.56 ± 0.13	0.23 ± 0.02	0.20 ± 0.02
KdM	68.9 ± 5.22	62.0 ± 4.70	77.6 ± 4.73	101 ± 5.92
KdM1	12.8 ± 2.01	21.1 ± 0.81	15.5 ± 0.54	21.3 ± 0.67
KdM2	41.3 ± 3.13	55.4 ± 1.70	46.5 ± 2.84	60.7 ± 3.55
KdM3	6.15 ± 0.41	7.32 ± 0.39	65.6 ± 5.09	52.4 ± 1.26

Mean value (*n* = 12) ± SD; M, Mc, MD, ME—extractable As concentration in the extraction solution (mg/kg); KdMi (i = M, M1, M2, M3)—distribution coefficient of As in the extraction solution (L/kg).

**Table 4 plants-12-03123-t004:** Values of the Cd concentrations in the CaCl_2_, DTPA, and EDTA extraction solutions and the values of the Kd distribution coefficients of Cd in these solutions.

Parameter (Cd)	S1	S2	S5	S6
M	0.07 ± 0.003	0.16 ± 0.005	0.05 ± 0.002	0.10 ± 0.002
M_C_	0.38 ± 0.01	0.64 ± 0.02	0.25 ± 0.02	0.79 ± 0.06
M_D_	0.21 ± 0.01	0.45 ± 0.02	0.19 ± 0.01	0.30 ± 0.03
M_E_	2.23 ± 0.07	3.88 ± 0.12	0.14 ± 0.12	0.28 ± 0.02
KdM	32.2 ± 3.28	28.7 ± 1.20	42.8 ± 1.66	29.5 ± 1.61
KdM1	6.67 ± 0.49	7.06 ± 0.34	8.65 ± 0.59	3.77 ± 0.24
KdM2	11.3 ± 1.15	10.1 ± 0.42	11.0 ± 0.47	10.0 ± 0.52
KdM3	1.04 ± 0.07	1.16 ± 0.07	15.0 ± 0.58	10.3 ± 0.56

Mean value (*n* = 12) ± SD; M, Mc, MD, ME—extractable Cd concentration in the extraction solution (mg/kg); KdMi (i = M, M1, M2, M3)—distribution coefficient of Cd in the extraction solution (L/kg).

**Table 5 plants-12-03123-t005:** The pH values of the extraction solutions.

Parameter	M_C_	M_D_	M_E_
pH *	2.60 ± 0.03	7.32 ± 0.28	7.10 ± 0.26

* pH unit; Mean value (*n* = 4) ± SD.

**Table 6 plants-12-03123-t006:** The values of the estimated cation exchange capacity (CEC) and soil organic carbon (SOC) in the studied soils.

Parameter (Cd)	CEC_M1	CEC_M2	CEC_M3	SOC
S1	27.9 ± 1.52	21.7 ± 1.98	38.2 ± 2.36	1.18 ± 0.16
S2	34.7 ± 2.09	22.0 ± 0.85	35.9 ± 3.52	1.21 ± 0.17
S3	29.0 ± 2.56	22.4 ± 1.66	38.5 ± 2.38	1.33 ± 0.22
S4	27.7 ± 1.62	22.4 ± 0.99	37.7 ± 1.26	1.15 ± 0.24
S5	30.9 ± 2.21	31.7 ± 2.12	39.8 ± 2.29	1.05 ± 0.11
S6	33.5 ± 1.62	28.1 ± 1.32	41.2 ± 3.06	6.27 ± 0.68

Mean value (*n* = 12) ± SD (standard error); CEC—cation exchange capacity in the extraction solution (meq/100 g).

**Table 7 plants-12-03123-t007:** The values of these two principal components (Component Loadings after Varimax Rotation) for As extraction solutions (M2, M2, M3) in soil S3.

Soil	Solution Extraction	The Values of Component Loadings after Varimax Rotation
S3	M1	PC1 = −0.324 pH_M1 + 0.642 SOC_s1 − 0.069 CEC_M1S3 − 0.122 AsS3 − 0.680 Kd_M1S3
		PC2 = 0.508 pH_M1 − 0.136 SOC_s1 − 0.447 CEC_M1S3 − 0.695 AsS3 − 0.200 Kd_M1S3
	M2	PC1 = 0.368 pH_M2 − 0.475 SOC_s1 − 0.449 CEC_M2S3 − 0.267 AsS3 − 0.604 Kd_M2S3
		PC2 = 0.457 pH_M2 + 0.165 SOC_s1 + 0.470 CEC_M2S3 − 0.727 AsS3 − 0.120 Kd_M2S3
	M3	PC1 = 0.572 pH_M3 − 0.309 SOC_s1 − 0.539 CEC_M3S3 − 0.102 AsS3 − 0.525 Kd_M3S3
		PC2 = 0.087 pH_M3 − 0.720 SOC_s1 + 0.069 CEC_M3S3 − 0.432 AsS3 + 0.531 Kd_M3S3

**Table 8 plants-12-03123-t008:** The values of these two principal components (Component Loadings after Varimax Rotation) for As extraction solutions (M2, M2, M3) in soil S4.

Soil	Solution Extraction	The Values of Component Loadings after Varimax Rotation
S4	M1	PC1 = −0.291 pH_M1 + 0.477 SOC_s1 − 0.280 CEC_M1S4 − 0.579 AsS4 − 0.592 Kd_M1S4
		PC2 = −0.005 pH_M1 − 0.334 SOC_s1 − 0.916 CEC_M1S4 − 0.172 AsS4 − 0.141 Kd_M1S4
	M2	PC1 = −0.127 pH_M2 + 0.605 SOC_s1 + 0.094 CEC_M2S4 − 0.647 AsS4 − 0.454 Kd_M2S4
		PC2 = −0.592 pH_M2 + 0.336 SOC_s1 − 0.530 CEC_M2S4 − 0.100 AsS4 − 0.120 Kd_M2S4
	M3	PC1 = −0.462 pH_M3 + 0.488 SOC_s1 − 0.053 CEC_M3S4 − 0.618 AsS4 − 0.403 Kd_M3S4
		PC2 = 0.366 pH_M3 − 0.147 SOC_s1 − 0.733 CEC_M3S4 − 0.127 AsS4 − 0.530 Kd_M3S4

**Table 9 plants-12-03123-t009:** The values of these two principal components (Component Loadings after Varimax Rotation) for As extraction solutions (M2, M2, M3) in soil S5.

Soil	Solution Extraction	The Values of Component Loadings after Varimax Rotation
S5	M1	PC1 = 0.598 pH_M1 − 0.414 SOC_s1 + 0.210 CEC_M1S5 + 0.142 AsS5 − 0.637 Kd_M1S5
		PC2 = 0.363 pH_M1 + 0.447 SOC_s1 − 0.419 CEC_M1S5 − 0.662 AsS5 − 0.235 Kd_M1S5
	M2	PC1 = −0.041 pH_M2 − 0.402 SOC_s1 − 0.469 CEC_M2S5 + 0.520 AsS5 + 0.587 Kd_M2S5
		PC2 = −0.896 pH_M2 + 0.289 SOC_s1 + 0.193 CEC_M2S5 + 0.272 AsS5 + 0.050 Kd_M2S5
	M3	PC1 = −0.498 pH_M3 + 0.340 SOC_s1 − 0.319 CEC_M3S5 − 0.511 AsS5 − 0.521 Kd_M3S5
		PC2 = −0.169 pH_M3 + 0.499 SOC_s1 + 0.800 CEC_M3S5 − 0.221 AsS5 + 0.207 Kd_M3S5

**Table 10 plants-12-03123-t010:** The values of these two principal components (Component Loadings after Varimax Rotation) for As extraction solutions (M2, M2, M3) in soil S6.

Soil	Solution Extraction	The Values of Component Loadings after Varimax Rotation
S6	M1	PC1 = −0.426 pH_M1 − 0.581 SOC_s2 + 0.407 CEC_M1S6 − 0.326 AsS6 + 0.457 Kd_M1S6
		PC2 = 0.634 pH_M1 + 0.173 SOC_s2 + 0.616 CEC_M1S6 + 0.183 AsS6 + 0.393 Kd_M1S6
	M2	PC1 = 0.433 pH_M2 + 0.594 SOC_s2 + 0.497 CEC_M2S6 + 0.332 AsS6 + 0.319 Kd_M2S6
		PC2 = −0.540 pH_M2 − 0.222 SOC_s2 + 0.450 CEC_M2S6 − 0.192 AsS6 + 0.647 Kd_M2S6
	M3	PC1 = 0.167 pH_M3 + 0.448 SOC_s2 − 0.697 CEC_M3S6 + 0.236 AsS6 + 0.478 Kd_M3S6
		PC2 = 0.545 pH_M3 − 0.498 SOC_s2 − 0.087 CEC_M3S6 − 0.538 AsS6 + 0.407 Kd_M3S6

**Table 11 plants-12-03123-t011:** The values of these two principal components (Component Loadings after Varimax Rotation) for Cd extraction solutions (M2, M2, M3) in soil S1.

Soil	Solution Extraction	The Values of Component Loadings after Varimax Rotation
S1	M1	PC1 = −0.440 pH_M1 − 0.513 SOC_s1 − 0.261 CEC_M1S1 − 0.935 CdS1 − 0.945 Kd_M1S1
		PC2 = −0.798 pH_M1 + 0.692 SOC_s1+ 0.388 CEC_M1S1 + 0.119 CdS1 + 0.073 Kd_M1S1
	M2	PC1 = 0.455 pH_M2 − 0.386 SOC_s1 − 0.275 CEC_M2S1 − 0.512 CdS1 − 0.551 Kd_M2S1
		PC2 = −0.306 pH_M2 + 0.095 SOC_s1 − 0.891 CEC_M2S1 + 0.288 CdS1 − 0.141Kd_M2S1
	M3	PC1 = 0.455 pH_M3 − 0.387 SOC_s1 + 0.272 CEC_M3S1 − 0.513 CdS1 − 0.552 Kd_M3S1
		PC2 = 0.300 pH_M3 − 0.090 SOC_s1 − 0.892 CEC_M3S1– 0.292 CdS1 + 0.142 Kd_M3S1

**Table 12 plants-12-03123-t012:** The values of these two principal components (Component Loadings after Varimax Rotation) for Cd extraction solutions (M2, M2, M3) in soil S2.

Soil	Solution Extraction	The Values of Component Loadings after Varimax Rotation
S2	M1	PC1 = 0.029 pH_M1 − 0.162 SOC_s1 + 0.473 CEC_M1S2 − 0.554 CdS2 − 0.663 Kd_M1S2
		PC2 = 0.59 pH_M1 − 0.695 SOC_s1 + 0.230 CEC_M1S2 + 0.310 CdS2 + 0.100 Kd_M1S2
	M2	PC1 = −0.365 pH_M2 + 0.396 SOC_s1 − 0.318 CEC_M2S2 − 0.427 CdS2 − 0.652 Kd_M2S2
		PC2 = −0.123 pH_M2 + 0.164 SOC_s1 − 0.696 CEC_M2S2 + 0.685 CdS2 + 0.059Kd_M2S2
	M3	PC1 = 0.577 pH_M3 − 0.245 SOC_s1 + 0.286 CEC_M3S2 − 0.432 CdS2 − 0.580 Kd_M3S2
		PC2 = −0.040 pH_M3 + 0.766 SOC_s1 − 0.379 CEC_M3S2 − 0.479 CdS2 − 0.194 Kd_M3S2

**Table 13 plants-12-03123-t013:** The values of these two principal components (Component Loadings after Varimax Rotation) for Cd extraction solutions (M2, M2, M3) in soil S5.

Soil	Solution Extraction	The Values of Component Loadings after Varimax Rotation
S5	M1	PC1 = 0.346 pH_M1 − 0.295 SOC_s1 − 0.508 CEC_M1S5 + 0.400 CdS5 − 0.611 Kd_M1S5
		PC2 = −0.633 pH_M1 − 0.102 SOC_s1 − 0.461 CEC_M1S5 − 0.544 CdS5 − 0.281 Kd_M1S5
	M2	PC1 = 0.655 pH_M2 + 0.019 SOC_s1 + 0.640 CEC_M2S5 + 0.985 CdS5 + 0.024 Kd_M2S5
		PC2 = 0.598 pH_M2 − 0.590 SOC_s1 − 0.687 CEC_ M2S5 − 0.066 CdS5 + 0.711 Kd_M2S5
	M3	PC1 = −0.644 pH_M3 + 0.272 SOC_s1 − 0.811 CEC_M3S5 + 0.935 CdS5 + 0.120 Kd_M3S5
		PC2 = 0.337 pH_M3 − 0.852 SOC_s1 + 0.146 CEC_M3S5 + 0.288 CdS5 + 0.931 Kd_M3S5

**Table 14 plants-12-03123-t014:** The values of these two principal components (Component Loadings after Varimax Rotation) for Cd extraction solutions (M2, M2, M3) in soil S6.

Soil	Solution Extraction	The Values of Component Loadings after Varimax Rotation
S6	M1	PC1 = −0.972 pH_M1 − 0.867 SOC_s2 + 0.128 CEC_M1S6 + 587 CdS6 + 0.501 Kd_M1S6
		PC2 = −0.117 pH_M1 + 0.447 SOC_s2 − 0.958 CEC_M1S6 + 0.159 CdS6 + 0.707 Kd_M1S6
	M2	PC1 = 0.904 pH_M2 + 0.996 SOC_s2 + 0.520 CEC_M2S6 − 0.316 CdS6 − 0.280 Kd_M2S6
		PC2 = 0.109 pH_M2 − 0.022 SOC_s2 − 0.036 CEC_M2S6 − 0.923 CdS6 − 0.913 Kd_M2S6
	M3	PC1 = −0.813 pH_M3 + 0.316 SOC_s2 + 0.176 CEC_M3S6 − 0.926 CdS6 − 0.942 Kd_M3S6
		PC2 = 0.194 pH_M3 + 0.837 SOC_s2 − 0.914 CEC_M3S6– 0.112 CdS6 − 0.266 Kd_M3S6

**Table 15 plants-12-03123-t015:** The values of As and Cd concentrations determined in the organs of the *Sinapis alba* grown in soils contaminated with As, Cd, and Ni.

Plant (*Sinapis alba*)	AsS5 mg/kg	AsS6 mg/kg	CdS5 mg/kg	CdS6 mg/kg
Root	2.56 ± 0.12	9.02 ± 0.12	0.44 ± 0.02	0.41 ± 0.05
Stem	0.75 ± 0.04	0.75 ± 0.04	0.29 ± 0.01	0.36 ± 0.04
Leaves	0.76 ± 0.05	2.18 ± 0.09	1.46 ± 0.10	2.52 ± 0.10
Flowers	0.75 ± 0.10	0.73 ± 0.09	0.11 ± 0.01	0.15 ± 0.02
Pods	2.32 ± 0.11	0.74 ± 0.02	0.18 ± 0.01	0.12 ± 0.01
Seeds	0.75 ± 0.06	0.74 ± 0.03	0.12 ± 0.02	0.10 ± 0.01
Plant	7.85 ± 0.25	14.2 ± 0.21	2.60 ± 0.14	3.66 ± 0.26

Mean value (*n* = 12) ± SD.

**Table 16 plants-12-03123-t016:** The values of the BAC-As and BAC-Cd determined in the organs of the *Sinapis alba* grown in soils contaminated with As, Cd, and Ni and the metal accumulation pattern.

Plant (*Sinapis alba*)	BAC-AsS5	BAC-AsS6	BAC-CdS5	BAC-CdS6
Root (R)	0.166 ± 0.006	0.532 ± 0.015	0.174 ± 0.003	0.132 ± 0.006
Stem (St)	0.048 ± 0.003	0.045 ± 0.003	0.126 ± 0.001	0.116 ± 0.005
Leaves (L)	0.049 ± 0.002	0.130 ± 0.001	0.671 ± 0.032	0.813 ± 0.024
Flowers (F)	0.047 ± 0.001	0.045 ± 0.003	0.048 ± 0.003	0.048 ± 0.003
Pods (P)	0.151 ± 0.006	0.044 ± 0.001	0.081 ± 0.005	0.039 ± 0.004
Seeds (Se)	0.048 ± 0.002	0.044 ± 0.002	0.052 ± 0.004	0.032 ± 0.002
Accumulation pattern	R > P > L > St, Se > F	R > L > St, F > P, Se	L > R > St > P > Se > F	L > R > St > F > P > Se

Mean value (*n* = 4) ± SD.

**Table 17 plants-12-03123-t017:** The values of these two principal components (Component Loadings after Varimax Rotation) for the variables included in the predictive model of metal concentration of As and Cd in *Sinapis alba*.

Variables	The Values of Component Loadings after Varimax Rotation (As)		The Values of Component Loadings after Varimax Rotation (Cd)	
	PC1	PC2	PC1	PC2
Kd_M1S5	−0.715	−0.142	0.132	0.763
Conc_plant_S5 *	0.933	−0.329	−0.084	0.931
Root_S5	0.680	−0.498	−0.642	0.347
Stem_S5	0.920	0.059	−0.687	0.059
Leaves_S5	0.886	0.088	0.467	0.812
Flowers_S5	0.039	−0.873	−0.909	−0.235
Pods_S5	0.657	0.149	−0.944	−0.169
Seeds_S5	0.051	−0.964	−0.848	−0.350

* Conc_plant_S5: total content of metal (As, Cd) in the plant according to experimental data.

**Table 18 plants-12-03123-t018:** The equations of the multiple regression model, The average experimental and predictive values of extractable As concentration in the organs of the *Sinapis alba,* and the values of the model validation parameters (aR^2^, DW, SW, and RMSE).

Soil	Plant (*Sinapis alba*)	The Equation of the Predictive Model *	aR^2^	As_exp_	As_pred_ ± SE	DW **	SW *	RMSE
S5	Root_S5	= 0.37 × Asplant_S5 − 0.003 × Kd_M1S5 × Asplant_S5	0.9985	2.56	2.54 ± 0.031	2.77	0.77	0.072
	Stem_S5	= 0.0007 × Asplant_S5 × Kd_M1S5 × Asplant_S5	0.9974	0.78	0.766 ± 0.062	2.27	0.80	0.039
	Leaves_S5	= −0.071 × Asplant_S5 + 0.012 × Asplant_S5 × Kd_M1S5	0.9967	0.75	0.761 ± 0.051	2.40	0.82	0.051
	Pods_S5	= −0.039 × Kd_M1S5 + 0.479 × Asplant_S5 − 0.0129 × (Asplant_S5)2	0.9912	2.32	2.317 ± 0.023	2.50	0.81	0.136
S6	Root_S6	= −0.039 × Kd_M1S6 × Kd_M1S6 + 0.088 × Asplant_S6 × Kd_M1S6	0.9989	9.03	9.06 ± 0.052	2.80	0.81	0.075
	Stem_S6	= 0.00012 × Kd_M1S6 × Asplant_S6 × Kd_M1S6	0.9976	0.75	0.754 ± 0.005	2.37	0.86	0.099
	Leaves_S6	= −0.053 × Kd_M1S5 + 0.21 × Asplant_S6	0.9966	2.76	2.79 ± 0.084	2.49	0.88	0.051
	Pods_S6	= 0.0002 × (Kd_M1S5 x Asplant_S6 × Asplant_S6)	0.9862	0.74	0.736 ± 0.006	2.88	0.88	0.028

* *p*-value < 0.05; ** (N = 12; k = 2).

**Table 19 plants-12-03123-t019:** The equations of the multiple regression model, the experimental and predictive values of extractable Cd concentrations in the leaves of the *Sinapis alba,* and the values of the model validation parameters (aR^2^, DW, SW, and RMSE).

Soil	Plant (*Sinapis alba*)	The Equation of the Predictive Model *	aR^2^	Cd_exp_	Cd_pred_ ± SE	DW **	SW *	RMSE
S5	Leaves_S5	= 0.16 × Kd_M1S5 + 0.21 × Cdplant_S5 − 0.014 * Kd_M1S5 * Cdplant_S5	0.9986	2.46	2.461 ± 0.012	2.29	0.86	0.093
S6	Leaves_S6	= 1.25 × Kd_M1S6 + 0.025 × (Cdplant_S6)2 − 0.19 × Cdplant_S6 × Kd_M1S6	0.9987	2.72	2.685 ± 0.055	2.36	0.87	0.102

* *p*-value < 0.05; ** (N = 12; k = 2); SE—standard error.

## Data Availability

Not applicable.

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
