# Peer review of "Evaluation of the Phytoremediation Potential of the Sinapis alba Plant Using Extractable Metal Concentrations"

_plants, 2023, doi:10.3390/plants12173123_

Round 1
Reviewer 1 Report
The manuscript assesses the effectiveness of using the Sinapis alba plant for soil phytoremediation, focusing on the bioaccumulation of As and Cd metals. The study identifies the factors that influence the extraction of metals from contaminated soils and develops predictive models for the plant’s potential as a phytoaccumulator or phytostabilizer for these metals. The study concludes that the Sinapis alba plant has a phytostabilizer potential for As and a phytoaccumulator potential for Cd.
Main issues identified in the manuscript:
The background section of the summary is too long and should be shortened. The results section of the summary needs to be expanded.
The aim of the study is not clearly stated in the introduction, and the wording needs to be revised for clarity.
The discussion section is too long and should be shortened.
The conclusion section should retain only one paragraph with 8-9 lines of content.
Recommendation: The reviewer suggests that the manuscript can be accepted after minor revisions addressing the issues mentioned above.
Author Response
Point 1: The background section of the summary is too long and should be shortened. The results section of the summary needs to be expanded.
Response 1:
We are grateful for the insightful and useful comments that we sought to implement in the revised version of the paper. We removed the graphs with the matrix of Pearson coefficients and replaced them with PCA analysis. We kept the RMS graphs, a PCA analysis was also added in section 2.6.1 (no. 261-273).
Point 2: The aim of the study is not clearly stated in the introduction, and the wording needs to be revised for clarity.
Response 2: We appreciate your feedback and agree and we made changes in the title and content.
Point 3: The discussion section is too long and should be shortened.
Response 3: We appreciate your feedback and, as consequance, the structure of the manuscript has been modified, except for the "Materials and methods" section.
Point 4: The conclusion section should retain only one paragraph with 8-9 lines of content.
Response 4: We appreciate your feedback and agree and the conclusion was revised.

Reviewer 2 Report
Authors claim to study the effects of extractable concentrations of arsenic (As) and cadmium (Cd) on the phytoremediation potential of White mustard. I honestly do not understand the objective of the authors' study. The approach seems incorrect or poorly described, and it needs to be corrected. The main issue is the lack of clarity regarding the relationship between the first part of the work and the second part. After several readings and a considerable effort, I might try to guess that the authors intend to use studies of different soil extractions with various extractants as input for a predictive model on how plants absorb metals from the soil???
However, this needs to be clearly stated, and the description of the entire paper should be presented step by step. For example, the authors should state their plan to create a model with X variables and explain how they established the first parameter of the model. Additionally, they should specify the factors that influence it the most. Currently, the paper lacks coherence.
Moreover, the authors need to justify why they are comparing chemical extraction, which aims to understand biological extraction, as there are significant differences between the two processes. For instance, plants actively uptake and transport nutrients and heavy metals through membrane transporters, and transpiration controls the movement of water and ions in plants. Furthermore, there are barriers that prevent the free travel of ions in plants, such as the endodermis.
If my guess was wrong, and the two parts of the study are not interrelated, the authors must explain why they are presented in a single paper.
Other comments:
Why there is no soil Cd/As extraction using just water as an extractor. Could watering the plants mobilize certain portions of arsenic and cadmium? Plants absorb nutrients from water, so that could be a simple explanation of how this model works. Conversely, have the authors considered the effect of plants on the soil, such as acidification, which is a common dicot method to increase nutrient uptake (if there is nutrient deficiency in soil – obviously Cd or As is not nutrient)?
There is no causal connection between many of the assumptions in the manuscript and the results and conclusions. In fact, most of the assumptions in the study are nonsensical. For example, the sentence "The values of Cd concentrations in the leaves of Sinapis alba indicate a process of accumulation in the roots and the transport of the metal in the leaves of the plant, behavior similar to plants grown in soil S5 as well as in soil S6" suggests that the authors claim that Cd concentration in leaves indicates the accumulation or transport of Cd in the roots. This is unclear.
Another issue arises with the sentence, "To evaluate the phytoaccumulator or phytostabilizer potential of Sinapis alba, As and Cd concentrations were analyzed in the root, stem, leaves, flowers, pods, and seeds of plants grown in soils contaminated with As, Cd, and Ni." Phytoaccumulation depends on the concentration of As/Cd in the plant, while phytostabilization depends on the stabilization of the metal in the roots, root surface, or rhizosphere due to the presence of plant exudates. Therefore, just having metal content we are not able to assess phytostability – HPLC of plant and soli could be suitable. Furthermore, the authors claim that Cd is stabilized in leaves. It is unclear how the authors distinguish stabilized Cd/As from accumulated Cd/As. They need to provide the methodology to clarify this.
The figures 1-8 with models are difficult to understand. It is unclear what the authors are trying to demonstrate. The authors need to carefully explain the types of dependencies shown in these graphs. For example, why are S3, S4, and S5 shown as parametric when they represent different soil types? Could the authors reduce the number of graphs and perhaps use a PCA (Principal Component Analysis) to highlight the main variables instead of 3D graphs?
Regarding text and figures 9 and 10, how was sorption and desorption estimates ? The tags in the figures appear to have been moved. Please correct them.
Once again, the authors must justify point by point how they predict the concentration of metals in plant organs based on soil properties. I do not believe there is a direct and predictable linear relationship. Such relationships may exist in the epidermis, especially within the root hair zone, but not further in the roots, stem, and leaves. Plants regulate their metal homeostasis and the short and long-distance transport of metals to various organs. As mentioned earlier, plants also interact with the soil to increase (via acidification) or decrease (via siderophores or nicotianamine exudation) ion uptake. The model should be simple and universal. However, in this case, each prediction has different coefficients and inputs. Furthermore, the distribution of metals depends on the metal accumulated in the plant, so the current approach does not appear to make sense.
At this point, I am unable to evaluate the merit of the work. The authors need to make it more straightforward, and the entire paper outline should be completely redone.
The syntax/grammar are terrible. I think that most of my problems with understanding the text is due to that. Authors should seek professional English editor help.
Author Response
Point 1: Authors claim to study the effects of extractable concentrations of arsenic (As) and cadmium (Cd) on the phytoremediation potential of White mustard. I honestly do not understand the objective of the authors' study. The approach seems incorrect or poorly described, and it needs to be corrected. The main issue is the lack of clarity regarding the relationship between the first part of the work and the second part. After several readings and a considerable effort, I might try to guess that the authors intend to use studies of different soil extractions with various extractants as input for a predictive model on how plants absorb metals from the soil???
Response 1: We are grateful for the insightful and useful comments that we sought to implement in the revised version of the paper.
The study of the effectiveness of the extraction solutions was carried out in order to determine a technique that best evaluates the metal concentration in soils contaminated with metals in high concentrations. We changed the title.
Point 2: However, this needs to be clearly stated, and the description of the entire paper should be presented step by step. For example, the authors should state their plan to create a model with X variables and explain how they established the first parameter of the model. Additionally, they should specify the factors that influence it the most. Currently, the paper lacks coherence.
Response 2: The methodology followed in the development of the regression equations, respectively, the parameters that were evaluated, can be found in "Materials and methods" (no. 645 - 690).
Point 3: Moreover, the authors need to justify why they are comparing chemical extraction, which aims to understand biological extraction, as there are significant differences between the two processes. For instance, plants actively uptake and transport nutrients and heavy metals through membrane transporters, and transpiration controls the movement of water and ions in plants. Furthermore, there are barriers that prevent the free travel of ions in plants, such as the endodermis.
If my guess was wrong, and the two parts of the study are not interrelated, the authors must explain why they are presented in a single paper.
Response 3: In the study, high metal concentrations were evaluated that could only be determined by chemical extractions. The study did not intend to present a methodology for the absorption of As and Cd metals in the Sinapis alba plant or the level of toxicity of these metals from a biological point of view. We have excluded from the content any interpretation that could refer to statements in this domain.
Point 4: Why there is no soil Cd/As extraction using just water as an extractor. Could watering the plants mobilize certain portions of arsenic and cadmium? Plants absorb nutrients from water, so that could be a simple explanation of how this model works. Conversely, have the authors considered the effect of plants on the soil, such as acidification, which is a common dicot method to increase nutrient uptake (if there is nutrient deficiency in soil – obviously Cd or As is not nutrient)?
Response 4: We appreciate your feedback and agree and we put the explanation in the text (no. 316 ÷ 319).
Point 5: There is no causal connection between many of the assumptions in the manuscript and the results and conclusions. In fact, most of the assumptions in the study are nonsensical. For example, the sentence "The values of Cd concentrations in the leaves of Sinapis alba indicate a process of accumulation in the roots and the transport of the metal in the leaves of the plant, behavior similar to plants grown in soil S5 as well as in soil S6" suggests that the authors claim that Cd concentration in leaves indicates the accumulation or transport of Cd in the roots. This is unclear.
Another issue arises with the sentence, "To evaluate the phytoaccumulator or phytostabilizer potential of Sinapis alba, As and Cd concentrations were analyzed in the root, stem, leaves, flowers, pods, and seeds of plants grown in soils contaminated with As, Cd, and Ni." Phytoaccumulation depends on the concentration of As/Cd in the plant, while phytostabilization depends on the stabilization of the metal in the roots, root surface, or rhizosphere due to the presence of plant exudates. Therefore, just having metal content we are not able to assess phytostability – HPLC of plant and soli could be suitable. Furthermore, the authors claim that Cd is stabilized in leaves. It is unclear how the authors distinguish stabilized Cd/As from accumulated Cd/As. They need to provide the methodology to clarify this.
Response 5: We appreciate your feedback and agree and we redid the presentation of the results of the study from the point of view of the accumulation of toxic metals in the plant.
Point 6: The figures 1-8 with models are difficult to understand. It is unclear what the authors are trying to demonstrate. The authors need to carefully explain the types of dependencies shown in these graphs. For example, why are S3, S4, and S5 shown as parametric when they represent different soil types? Could the authors reduce the number of graphs and perhaps use a PCA (Principal Component Analysis) to highlight the main variables instead of 3D graphs?
Response 6: We appreciate your feedback and agree and we extracted the graphs of the Pearson correlation coefficient matrix. These were replaced with the results obtained in the PCA analysis. The graphs of the response surface were redone according to these results. The response surface diagrams are useful to show the time evolution and the interactions between the parameters that can influence the metal desorption process in the analyzed solutions.
Point 7: Regarding text and figures 9 and 10, how was sorption and desorption estimates ? The tags in the figures appear to have been moved. Please correct them.
Response 7: They have been corrected in the figure 9 and figure 10.

Round 2
Reviewer 2 Report
Problem 1:
Line 24: "…> Seeds, stems > Flowers for ca and leaves >…" What does "ca" stand for?
Line 38-39: "Sources of soil contamination with toxic metals must significantly increase the plant-available metal concentration in the soil to be detectable" – this is nonsensical. Please use short, simple sentences as currently the text is impossible to understand. All phrases from 38-42 are again confusing…
"Remediation of contaminated soils is largely determined by metal concentrations and the form in which they exist. Sources of soil contamination with toxic metals must significantly increase the plant-available metal concentration in the soil to be detectable. The most important factor governing the phyto-availability of toxic metals in soil is the solubility of the metal associated with the solid phase [4 - 6].”
It seems that the paragraph starts generally about remediation and then shifts to discussing phytoavailability, or the authors mistakenly use these words as synonyms. However, there is no clear connection between these parts. As a reader, I shouldn't have to guess what the authors want to convey.
Next, authors say: "The rate of release of soluble species strongly influences the rate and degree of uptake, mobility, and toxicity of the metal in plants and consuming animals."
1. What is the difference between the rate and degree of uptake?,
2. Uptake to the symplast depends on metal transporters that have low specificity for the target nutrient and take up the toxic ion – therefore there is no simple relation between the rate of release of soluble species and plant metal uptake – there may be a correlation, but it is not as straightforward as the sentence implies.
3. How does soil solubility influence metal toxicity in plants? The toxicity of the metal depends on its chemical properties, e.g., oxidative state. There are some relations between oxidative state, mobility, and toxicity of elements e.g., for As (III) and As (V) but not for Cd… which most often has an oxidation state +2 and its mobilized by acidic conditions (it stays in solution at pH up to 6.5 -> one of the highest compares to other common metals in soil), presence of other ions (e.g. specific cations and anions) or presence or absence of specific organic matter and so forth… There are complex relationships, and in scientific papers, they cannot be put in general statements like that. Authors need to introduce As and Cd properties separately.
4. It is not possible to correlate directly the rate of soluble species in the soil in the same way between plant and animal consuming plant – e.g., Sinapis alba, as authors show, accumulate more As in roots and more Cd in shoots. Animals can consume roots and/or shoots and depending on their diet will take up either a lot more (than soil level) or much less (trace level) than what is soluble in the soil – this is the wrong causality.
...from the further part, different example, Line 225 – title "Evaluation of the efficacity of As and Cd in M1-CaCl2, M2-DTPA, and M3-EDTA solutions" – the word is efficacy (also line 76) and this sentence now means that there is a study to assess the effectiveness (efficacy) of two substances: As (Arsenic) and Cd (Cadmium), in three different solutions: M1-CaCl2, M2-DTPA, and M3-EDTA – I am sure that is not what the authors are doing in this part…
Concluding remark 1: Please use a professional English editing service with the option of a scientific editor that will help with all the English and logical issues within the text – this is not a role for the reviewers. I did encounter more issues within the text but have no patience to correct or highlight...
Problem 2:
Sentence line 318-321: "The aqueous extracts were not used in this study, because the water used as an extraction solution extracts soluble forms of the chemical species of As and Cd that do not reflect the real level of contamination of the analyzed soils."
In this part, authors explain the lack of using water for extraction because it extracts soluble forms of As and Cd not showing the not soluble levels (total level) – but how authors know that?
Comparing EDTA/DTPA/CaCl2 methods to water would show relative differences that are important for assessing efficiency of the method. Authors claim in another sentence that they look for the extractant that has the most efficiency (that means the best effect with the lowest cost). The most frustrating is the fact that authors justify the lack of aqueous extraction using reference [37 - Mondragón-Solórzano et al 2016] that describe extraction from a water solution (sic!)…
However, in earlier part, authors justify the idea of using their extraction solutions with reference [36 -> Moghal et al 2020] that clearly states the art: “Four molar concentrations (0.01, 0.1, 0.5, and 1.0 M) and a solid to liquid ratio of 1:20 were selected for the desorption testing. Apart from these four extractants, double distilled water (DI) was also used for comparison purposes” (Moghal et al 2020).
Concluding remark 2: The extraction experiments and analysis must be repeated using, apart from three extractants, a DI water (as a forth extractant) for comparison reasons.
Problem 3:
Lastly, the Point 3 and Response 3 (authors' response) - I will rephrase the question: What the first part of the research (extraction of As/Cd from the soil using M1-3) [part 1 of the manuscript] have to do with the phytoremediation potential of Sinapis alba [part 2 of the manuscript]? How are these two different studies connected? What data from part 1 are used in part 2?
The answers for the above questions are crucial for readers as it will make the study more comprehensive, and the motivation behind it will be clear. Using those answers authors should craft linking between parts of the manuscript: 1) in the abstract and 2) in the appropriate transition place from one part to the other. I still lack clear connection between part 1 and part 2, although in some places, there are some explanations, those do not make a proper connection.
Minor:
Line 503 "The results of the PCA analysis for Cd showed that the second main component" should be the first main component, not second.
Additionally, please make a highlight of all the changes between the versions of the manuscript highlighted – in the new version of the manuscript, only part of the changes is highlighted; others are not.
Done in the main comment but well agian please use a professional English editing service with the option of a scientific editor that will help with all the English and logical issues within the text – this is not a role for the reviewers. I did encounter more issues within the text but have no patience to correct or highlight...
Author Response
Response to Reviewer 2 Comments
Round 2
Point 1:
- What is the difference between the rate and degree of uptake?,
- Uptake to the symplast depends on metal transporters that have low specificity for the target nutrient and take up the toxic ion – therefore there is no simple relation between the rate of release of soluble species and plant metal uptake – there may be a correlation, but it is not as straightforward as the sentence implies.
- How does soil solubility influence metal toxicity in plants? The toxicity of the metal depends on its chemical properties, e.g., oxidative state. There are some relations between oxidative state, mobility, and toxicity of elements e.g., for As (III) and As (V) but not for Cd… which most often has an oxidation state +2 and its mobilized by acidic conditions (it stays in solution at pH up to 6.5 -> one of the highest compares to other common metals in soil), presence of other ions (e.g. specific cations and anions) or presence or absence of specific organic matter and so forth… There are complex relationships, and in scientific papers, they cannot be put in general statements like that. Authors need to introduce As and Cd properties separately.
- It is not possible to correlate directly the rate of soluble species in the soil in the same way between plant and animal consuming plant – e.g., Sinapis alba, as authors show, accumulate more As in roots and more Cd in shoots. Animals can consume roots and/or shoots and depending on their diet will take up either a lot more (than soil level) or much less (trace level) than what is soluble in the soil – this is the wrong causality.
...from the further part, different example, Line 225 – title "Evaluation of the efficacity of As and Cd in M1-CaCl2, M2-DTPA, and M3-EDTA solutions" – the word is efficacy (also line 76) and this sentence now means that there is a study to assess the effectiveness (efficacy) of two substances: As (Arsenic) and Cd (Cadmium), in three different solutions: M1-CaCl2, M2-DTPA, and M3-EDTA – I am sure that is not what the authors are doing in this part…
Concluding remark 1: Please use a professional English editing service with the option of a scientific editor that will help with all the English and logical issues within the text – this is not a role for the reviewers. I did encounter more issues within the text but have no patience to correct or highlight...
Response 1: We are grateful for the insightful and useful comments that we sought to implement in the revised version of the paper. In the Introduction section, in agreement with the instructions of paragraphs 1- 4, we have changed the content from lines 37-61 and lines 75-82. Translation and phrases have been attentively evaluated.
Point 2: Sentence line 318-321: "The aqueous extracts were not used in this study, because the water used as an extraction solution extracts soluble forms of the chemical species of As and Cd that do not reflect the real level of contamination of the analyzed soils."
In this part, authors explain the lack of using water for extraction because it extracts soluble forms of As and Cd not showing the not soluble levels (total level) – but how authors know that?
Comparing EDTA/DTPA/CaCl2 methods to water would show relative differences that are important for assessing efficiency of the method. Authors claim in another sentence that they look for the extractant that has the most efficiency (that means the best effect with the lowest cost). The most frustrating is the fact that authors justify the lack of aqueous extraction using reference [37 - Mondragón-Solórzano et al 2016] that describe extraction from a water solution (sic!)…
However, in earlier part, authors justify the idea of using their extraction solutions with reference [36 -> Moghal et al 2020] that clearly states the art: “Four molar concentrations (0.01, 0.1, 0.5, and 1.0 M) and a solid to liquid ratio of 1:20 were selected for the desorption testing. Apart from these four extractants, double distilled water (DI) was also used for comparison purposes” (Moghal et al 2020).
Concluding remark 2: The extraction experiments and analysis must be repeated using, apart from three extractants, a DI water (as a forth extractant) for comparison reasons.
Response 2: We appreciate your feedback and agree with these changes. Following the instructions, we completed a study with the method of water extraction, in 2.4 section. Although experimental results were initially collected, we did not consider them representative of the completion of this work, especially in the case of the extraction of Cd from the soils studied. I think the reported results confirm our decision.
Point 3: Lastly, the Point 3 and Response 3 (authors' response) - I will rephrase the question: What the first part of the research (extraction of As/Cd from the soil using M1-3) [part 1 of the manuscript] have to do with the phytoremediation potential of Sinapis alba [part 2 of the manuscript]? How are these two different studies connected? What data from part 1 are used in part 2?
The answers for the above questions are crucial for readers as it will make the study more comprehensive, and the motivation behind it will be clear. Using those answers authors should craft linking between parts of the manuscript: 1) in the abstract and 2) in the appropriate transition place from one part to the other. I still lack clear connection between part 1 and part 2, although in some places, there are some explanations, those do not make a proper connection.
Response 3: We appreciate your comments. In this context, we have included in Abstract and Section 3.4 the content that we considered to satisfy your requirements. The content of the conclusions has been improved.
Point 4: Line 503 "The results of the PCA analysis for Cd showed that the second main component" should be the first main component, not second.
Response 4: Done for line 531 (modified line 503).
Thanking for your kind assistance,
Yours sincerely,
Drd. Eng. Nicoleta Vasilache

Round 3
Reviewer 2 Report
Thank you.
The professional English editing is required.